# Generation Order and Parallel Decoding in Masked Diffusion Models: An Information-Theoretic Perspective

## Abstract

Masked Diffusion Models (MDMs) significantly accelerate inference by trading off sequential determinism. However, the theoretical mechanisms governing generation order and the risks inherent in parallelization remain under-explored. In this work, we provide a unified information-theoretic framework to decouple and analyze two fundamental sources of failure: *order sensitivity* and *parallelization bias*. Our analysis yields three key insights: (1) The benefits of *Easy-First* decoding (prioritizing low-entropy tokens) are magnified as model error increases; (2) factorized parallel decoding introduces intrinsic sampling errors that can lead to arbitrary large Reverse KL divergence, capturing "incoherence" failures that standard Forward KL metrics overlook; and (3) while verification can eliminate sampling error, it incurs an exponential cost governed by the total correlation within a block. Conversely, heuristics like remasking, though computationally efficient, cannot guarantee distributional correctness. Experiments on a controlled Block-HMM and large-scale MDMs (LLaDA) for arithmetic reasoning validate our theoretical framework.

## 1. Introduction

Autoregressive (AR) models have established themselves as the dominant paradigm for discrete sequence generation, powering breakthroughs in natural language processing (Achiam et al., 2023; Touvron et al., 2023) and biological sequence design (Madani et al., 2023). By decomposing the joint probability of a sequence into a chain of conditional probabilities, these models allow for tractable training and high-fidelity generation. However, this factorization imposes a strict sequential constraint: standard decoding proceeds token-by-token from left to right, resulting in inference latency that scales linearly with sequence length.

To dismantle this computational bottleneck, recent research has pivoted toward parallel generation strategies. An initial line of work focuses on *Speculative Decoding* (Leviathan et al., 2023; Chen et al., 2023). However, the efficiency of speculative methods is strictly capped by the *draft acceptance rate*, which often stagnates in complex reasoning tasks or when the draft-oracle gap is large (Wen et al., 2024).

To achieve more aggressive speedups, a more radical shift has emerged toward fully parallel and non-autoregressive frameworks, such as *Masked Diffusion Models (MDMs)* (Sahoo et al., 2024; Nie et al., 2025) and *Discrete Flow Matching* (Campbell et al., 2024). While these models bypass the serial verification bottleneck of speculative decoding, they introduce two unaddressed theoretical risks: (1) **Order Sensitivity**: Due to model approximation errors, the magnitude of error accumulation becomes highly dependent on the generation order. (2) **Parallelization Bias**: parallelization itself introduces a structural error; even with a perfect model, the resulting decoding distribution diverges from the ground truth. This leads to the generation of individually likely but jointly impossible sequences, a phenomenon we term incoherence or hallucinations.

A variety of heuristic solutions have been proposed, such as confidence-based masking (Chang et al., 2022), entropy-based scheduling (Peng et al., 2025), and margin-based methods (Kim et al., 2025). These approaches typically rely on what we term the **Easy-first principle**—prioritizing the generation of tokens with low uncertainty. However, the field lacks a unified theoretical framework to separate and characterize the distinct sources of failure in these strategies, especially as they typically intertwine order scheduling with parallel decoding. In particular, two fundamental questions remain open: *Can the widely-used "Easy-first" principle be justified theoretically?* and *To what extent can the errors induced by parallel decoding be mitigated or eliminated?*.

**Contributions.** This work provides an information-theoretic analysis of how generation order and parallelization affect distributional correctness in MDMs. Our main contributions are:

[1]Anonymous Institution, Anonymous City, Anonymous Region, Anonymous Country. Correspondence to: Anonymous Author <anon.email@domain.com>.

Preliminary work. Under review by the International Conference on Machine Learning (ICML). Do not distribute.

- **Generation order matters under model error.** We show that under imperfect conditional models, the divergence between the induced and target joint distributions is explicitly order-dependent due to rollout-induced covariate shift. Our analysis indicates that Easy-First ordering yields greater benefits as model error increases, due to the amplified effect of early conditional inaccuracies under forward KL divergence.

- **Parallel decoding introduces intrinsic sampling error.** We compare forward KL, reverse KL, and incoherence probability as metrics for evaluating parallel decoding. We show that sampling error in factorized parallel decoding is unavoidable: exact elimination requires verification with expected cost exponential in forward KL, while heuristic corrections, though computationally efficient, in general cannot guarantee distributional correctness and totally avoid incoherence.

- **Empirical validation**. Experiments on a fully specified Block-HMM confirm that parallel mean-field decoding can exhibit high reverse KL and incoherence despite modest forward KL. Additional results on arithmetic reasoning and masked diffusion decoding demonstrate that Easy-First schedules substantially improve accuracy, especially in high-error regimes.

## 2. Background and Problem Setup

### 2.1. Autoregressive Models and Decoding

We consider a length-$T$ discrete sequence $X_{1:T} = (X_1, \ldots, X_T)$ over a finite alphabet $\mathcal{V}$, distributed according to an unknown data distribution $p_{\text{data}}$. Autoregressive (AR) models provide a convenient reference point by parameterizing a joint distribution via the chain rule:

$$p_\theta(x_{1:T}) = \prod_{t=1}^{T} p_\theta(x_t \mid x_{<t}), \qquad (1)$$

where $x_{<t} := (x_1, \ldots, x_{t-1})$ denotes the realized prefix.

Decoding refers to the procedure used to sample from the model conditionals. While standard AR decoding is sequential, recent methods aim to reduce latency by generating multiple variables in parallel. A common abstraction is *blockwise decoding*, which partitions positions into blocks and generates all tokens in a block simultaneously, conditioned on tokens generated in previous blocks. Depending on the algorithm, intra-block dependencies may be ignored, approximated, or corrected through additional steps such as accept–reject or iterative refinement.

Our analysis concerns the distributional consequences of decoding strategies rather than model architectures. The same trained model can induce different output distributions under different decoding procedures.

### 2.2. Error Sources and Scope

We distinguish two sources of generation error: model error ($p_\theta \approx p_{\text{data}}$) and sampling error (mismatch introduced by parallel or approximate decoding). We analyze how generation order and parallelization affect these errors to optimize speed–fidelity trade-offs. This analysis applies formally to autoregressive and AR-compatible masked diffusion models (e.g., LLaDA (Nie et al., 2025)) that admit explicit conditional decomposition. For general diffusion or flow-based models without such decomposition, our findings serve as qualitative guidance.

## 3. When Do Order and Parallelization Matter?

This section formalizes how generation schemes influence the target distribution. We identify two idealized regimes where generation order and parallelization remain mathematically invariant. The divergence between these ideals and realistic distributions motivates our subsequent analysis of order-sensitive and parallelization-sensitive strategies.

Figure 1 provides a minimal illustration of this distinction. Under idealized assumptions, different generation orders or factorizations induce identical joint distributions. In contrast, under model error or conditional dependence, discrepancies compound along the generation path, leading to order- and factorization-dependent outcomes.

### 3.1. Idealized Regime I: Perfect Modeling ⇒ Order-Invariance

Consider an oracle setting where the true data conditionals of $p_{\text{data}}$ are accessible and generation is strictly sequential. For any permutation $\pi$ of $\{1, \ldots, T\}$, the chain rule yields the equivalent factorization

$$p_{\text{data}}(x_{1:T}) = \prod_{t=1}^{T} p_{\text{data}}\big(x_{\pi(t)} \mid x_{\pi(<t)}\big), \qquad (2)$$

where $x_{\pi(<t)} := (x_{\pi(1)}, \ldots, x_{\pi(t-1)})$ denotes the realized context generated so far (and $X_{\pi(<t)}$ its random counterpart).

In this regime, sampling sequentially according to the exact conditional $x_{\pi(t)} \sim p_{\text{data}}(\cdot \mid x_{\pi(<t)})$ produces a valid draw from the same joint distribution $p_{\text{data}}(x_{1:T})$ for *any* choice of $\pi$. Consequently, the induced output distribution is order-invariant: left-to-right and right-to-left decoding are distributionally equivalent under perfect modeling and strictly sequential sampling.

> **Implication.** If $p_\theta = p_{\text{data}}$ and decoding is strictly sequential (one variable at a time using exact conditionals), then the induced joint distribution over $X_{1:T}$ is invariant to the generation order $\pi$.

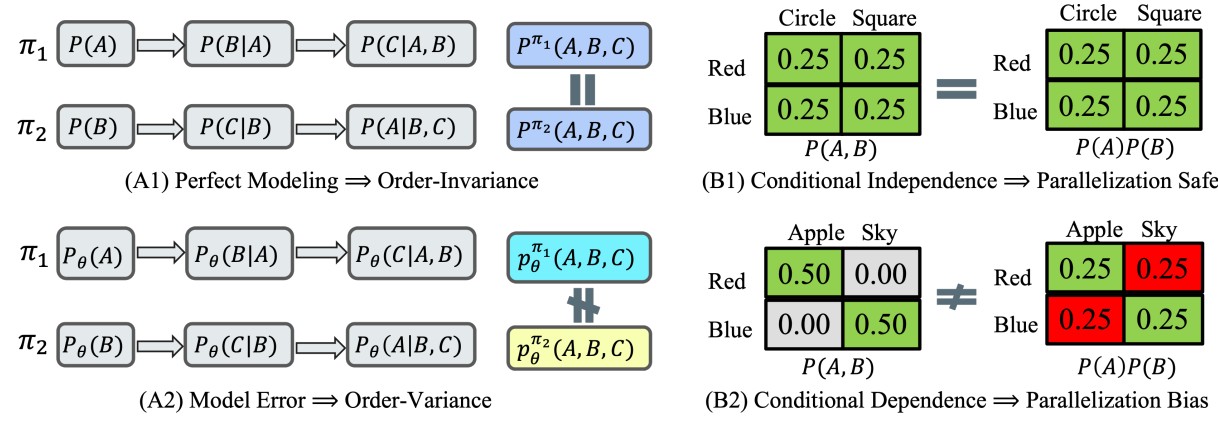

Figure 1. Order sensitivity and parallelization bias under imperfect factorization. Minimal examples illustrating that different generation orders $(\pi_1, \pi_2)$ or parallel factorizations induce identical joint distributions under ideal conditions, but diverge under model error or conditional dependence.

### 3.2. Realistic Regime I: Model Error and the Uncertainty Proxy

In practice, we rely on an approximated model $p_\theta(\cdot \mid \cdot)$. Even with sequential decoding, prediction errors accumulate along the generated path. Crucially, the *divergence accumulated along decoding trajectories* is **path-dependent**, because local errors are evaluated under prefix distributions induced by the model itself. Consequently, unlike the idealized regime where all generation paths are equivalent, in the realistic regime, different generation orders $\pi$ accumulate total error differently. This motivates the need to identify generation orders that minimize cumulative divergence, as we analyze formally in Sec. 4.

### 3.3. Idealized Regime II: Conditional Independence ⇒ Exact Parallelization

To isolate the effect of decoding from model approximation, we assume oracle access to a target joint distribution $p(x_{1:T})$. Our focus is not density estimation, but the *distributional bias induced by parallel decoding*.

Let $\mathcal{B} = \{B_m\}_{m=1}^M$ be a partition of $\{1, \ldots, T\}$ into decoding blocks, and define

$$P_m := \bigcup_{\ell < m} B_\ell \qquad (3)$$

as the set of variables generated prior to block $B_m$. A blockwise decoding algorithm induces the joint distribution

$$p_{\mathcal{B}}(x_{1:T}) = \prod_{m=1}^M p_{\prod}(x_{B_m} \mid x_{P_m}), \qquad (4)$$

where $p_{\prod}(\cdot \mid x_{P_m})$ is the proposal used to generate block $B_m$.

Parallel decoding typically employs a mean-field approximation within each block,

$$p_{\prod}(x_{B_m} \mid x_{P_m}) = \prod_{i \in B_m} p(x_i \mid x_{P_m}), \qquad (5)$$

which preserves correct marginal conditionals but discards intra-block dependencies present in $p(x_{B_m} \mid x_{P_m})$.

Under this condition, the conditional joint distribution over the block factorizes completely. Consequently, sampling all $x_i \in B$ in parallel from their conditionals is equivalent to any sequential sampling order within the block. In this idealized regime, parallel decoding induces the same joint distribution over $X_{1:T}$ as strictly sequential generation.

> **Implication.** If block variables are conditionally independent given the context, parallel generation induces no sampling bias and is distributionally equivalent to sequential decoding.

### 3.4. Realistic Regime II: Conditional Dependence ⇒ Factorization Bias

Real-world structured sequences—such as natural language, code, or symbolic reasoning traces—rarely satisfy the conditional independence assumption in Eq. (5) for nontrivial blocks. Instead, strong intra-block dependencies are the norm, arising from syntactic agreement, semantic coherence, or logical constraints.

Unlike the sequential case—where discrepancies arise from *model error* in approximating $p_{\text{data}}$—parallel decoding introduces a distinct source of error: **sampling bias** due to mismatched factorization. Crucially, this bias persists even when all marginal conditional distributions are exact, as long as intra-block dependencies are ignored at sampling time. As a result, the distribution induced by parallel decoding

deviates from the target distribution, motivating a formal analysis of sampling error in Sec. 5.

## 4. Model Error and Order-Sensitive Generation

### 4.1. Sequential Generation Under Model Error

Consider a learned model $p_\theta^\pi(x_{1:T}) = \prod_{t=1}^{T} p_\theta(x_{\pi(t)} \mid x_{\pi(<t)})$ approximating the data distribution $p_{\text{data}}$. During decoding, the model conditions on its own previously generated outputs. As a result, conditional errors are evaluated under the evolving prefix distribution $p_\theta(x_{\pi(<t)})$ rather than under $p_{\text{data}}$.

For a prefix $h = x_{\pi(<t)}$, define the local conditional error

$$\epsilon_\theta(h) := \text{KL}\big(p_\theta(\cdot \mid h) \,\big\|\, p_{\text{data}}(\cdot \mid h)\big). \quad (6)$$

**Lemma 4.1** (Rollout-weighted KL decomposition). *For any permutation $\pi$,*

$$\text{KL}\big(p_\theta^\pi \,\big\|\, p_{\text{data}}\big) = \sum_{t=1}^{T} \mathbb{E}_{x_{\pi(<t)} \sim p_\theta^\pi}\big[\epsilon_\theta\big(x_{\pi(<t)}\big)\big]. \quad (7)$$

Lemma 4.1 makes the order dependence explicit: although the algebraic chain rule is permutation invariant, the expectation in Eq. (7) is taken under the order-dependent rollout $p_\theta^\pi(x_{\pi(<t)})$. Errors made early in the sequence therefore shift the prefix distribution and affect all subsequent conditional divergences.

### 4.2. Objective and Error Amplification

Our objective is to select a generation order that minimizes the divergence induced by autoregressive rollout under an imperfect model:

$$\pi^\star \in \arg\min_{\pi} \; \text{KL}\big(p_\theta^\pi \,\big\|\, p_{\text{data}}\big), \quad (8)$$

$$= \arg\min_{\pi} \; \sum_{t=1}^{T} \mathbb{E}_{x_{\pi(<t)} \sim p_\theta^\pi}\big[\epsilon_\theta\big(x_{\pi(<t)}\big)\big], \quad (9)$$

where the equality follows from Lemma 4.1.

Direct optimization of Eq. (9) is intractable due to the combinatorial space of permutations and the dependence of each expectation on the order-induced rollout distribution. To enable principled comparison between generation orders, we introduce the following assumption, motivated by empirical observations.

**Assumption 4.2** (Entropy-dominated local error). For prefixes encountered during autoregressive rollout under order $\pi$, the expected local approximation error at step $t$ satisfies

$$\bar{\epsilon}_{\pi(t)} \leq \alpha\, H\big(X_{\pi(t)} \mid x_{\pi(<t)}\big) + \lambda\, \bar{\epsilon}_{\pi(<t)} + b, \quad (10)$$

where $\bar{\epsilon}_{\pi(t)} := \mathbb{E}_{x_{\pi(<t)} \sim p_\theta^\pi}[\epsilon_\theta(x_{\pi(<t)})]$, $\alpha > 0$ captures the contribution of intrinsic uncertainty, $\lambda \geq 0$ models prefix-induced error propagation, and $b \geq 0$ absorbs residual mismatch.

This assumption abstracts the empirically observed relationship between conditional entropy and approximation error, while allowing for mild prefix-dependent effects. In App. B, we provide empirical evidence that conditional entropy is a strong predictor of approximation error, and that prefix error exhibits a monotone but secondary influence.

Substituting Eq. (10) into Eq. (9) yields an upper bound of the form

$$\mathcal{J}(\pi) \leq \sum_{t=1}^{T} W_t\, H\big(X_{\pi(t)} \mid x_{\pi(<t)}\big) + \tilde{b}, \quad (11)$$

where $W_t = \alpha(1 + \lambda)^{T-t}$ and $\tilde{b}$ is a constant independent of $\pi$.

### 4.3. Easy-First as a Reliable Approximation

**Effect of model error.** The bound in Eq. (11) makes the role of model error explicit. As prefix sensitivity becomes stronger, the induced weight sequence places increasing emphasis on uncertainty resolved at earlier steps. Under such conditions, deviations from entropy-minimizing orders incur larger penalties, making Easy-First a more *risk-averse* approximation that limits worst-case error amplification under prefix shift. At generation step $t$, the greedy (Easy-first) policy select the next variable with minimal immediate conditional entropy among the remaining variables:

$$\pi(t) = \arg\min_{j \notin \pi(<t)} H(X_j \mid x_{\pi(<t)}). \quad (12)$$

In practice, exact conditional entropies can be approximated using efficient uncertainty surrogates, such as model confidence (Nie et al., 2025) or probability margins (Kim et al., 2025). The analysis relies only on the assumption that higher intrinsic uncertainty correlates with larger local approximation error.

> **Implication.** Easy-First ordering yields greater benefits as model error increases, due to the amplified effect of early conditional inaccuracies under forward KL.

## 5. Sampling Error Induced by Parallel Generation

### 5.1. Three Metrics for Sampling Error

Recent theoretical work on masked diffusion inference schedules primarily measures degradation via expected divergence between the true and sampled distributions (e.g.,

forward-KL-type criteria) (Chen et al., 2025), whereas ParallelBench demonstrates that parallel decoding can fail dramatically on dependency-sensitive tasks that standard benchmarks miss (Kang et al., 2025). Motivated by this gap, we distinguish forward KL, reverse KL, and incoherence as complementary metrics capturing information loss, sampling risk, and catastrophic failure frequency.

**Forward KL (information loss).** A common choice in prior work is the forward KL divergence $\mathrm{KL}(p \,\|\, p_{\mathcal{B}})$, which measures how well the decoding distribution $p_{\mathcal{B}}$ approximates the target distribution $p$ in expectation under $p$. This metric is well suited for characterizing information loss and average dependence distortion, and is closely related to notions such as (conditional) total correlation. However, because expectations are taken under $p$, forward KL is largely insensitive to configurations that are rare or absent under $p$ but assigned non-negligible probability by $p_{\mathcal{B}}$.

**Reverse KL (sampling risk).** In contrast, reverse KL evaluates the quality of samples actually produced by the decoding procedure. Samples $x \sim p_{\mathcal{B}}$ are scored under the target distribution $p$ via the expected negative log-likelihood

$$\mathcal{R}(p_{\mathcal{B}}; p) := \mathbb{E}_{x \sim p_{\mathcal{B}}}\big[ -\log p(x)\big], \qquad (13)$$

which differs from $\mathrm{KL}(p_{\mathcal{B}}\|p)$ only by the entropy of $p_{\mathcal{B}}$. Because expectations are taken under the sampling distribution, reverse KL directly penalizes decoding-time failures, including configurations that violate semantic or structural constraints. As a result, reverse KL is explicitly sensitive to *support mismatch* and provides an operational notion of sampling error.

**Incoherence (event-level failure).** Beyond distributional divergences, one may also consider the probability that a decoded sample is incoherent or implausible under the target distribution, e.g.,

$$\Pr_{x \sim p_{\mathcal{B}}}\big[p(x) \leq \epsilon\big], \qquad (14)$$

for a small threshold $\epsilon$. This metric captures the frequency of catastrophic decoding failures and is particularly aligned with practical notions of hallucination. However, incoherence is threshold-dependent and does not capture the full distributional severity of errors; it is best viewed as a diagnostic rather than a complete metric.

**Relationship and choice.** These three metrics emphasize different aspects of parallel decoding error. Forward KL reflects average information loss under the target distribution, incoherence measures the frequency of extreme failures, and reverse KL provides a unified distributional risk that penalizes both the frequency and severity of implausible samples.

## 5.2. Forward vs. Reverse KL through Conditional Total Correlation

Both forward and reverse KL divergences admit natural decompositions in terms of blockwise conditional dependencies. These decompositions clarify the distinct aspects of parallel decoding error captured by each divergence.

**Forward KL and conditional total correlation.** We first define the conditional total correlation within block $B_m$ given $x_{P_m}$ as

$$\overrightarrow{\mathcal{TC}}_m(x_{P_m}) = \mathrm{KL}\big(p(x_{B_m} \mid x_{P_m}) \,\big\|\, p_{\prod}(x_{B_m} \mid x_{P_m})\big), \qquad (15)$$

where $p_{\prod}(x_{B_m} \mid x_{P_m}) = \prod_{i \in B_m} p(x_i \mid x_{P_m})$ as in Eq. (5). Applying the chain rule to the forward KL divergence yields

$$\mathrm{KL}(p \,\|\, p_{\mathcal{B}}) = \sum_{m=1}^{M} \mathbb{E}_{x_{P_m} \sim p}\Big[\overrightarrow{\mathcal{TC}}_m(x_{P_m})\Big]. \qquad (16)$$

This quantity measures the *strength of intra-block dependence* under the true distribution and characterizes the information loss incurred by ignoring these dependencies on average under $p$.

**Reverse KL and reverse conditional total correlation.** Analogously, define the reverse conditional total correlation as

$$\overleftarrow{\mathcal{TC}}_m(x_{P_m}) = \mathrm{KL}\big(p_{\prod}(x_{B_m} \mid x_{P_m}) \,\big\|\, p(x_{B_m} \mid x_{P_m})\big). \qquad (17)$$

The reverse KL divergence then decomposes as

$$\mathrm{KL}(p_{\mathcal{B}} \,\|\, p) = \sum_{m=1}^{M} \mathbb{E}_{x_{P_m} \sim p_{\mathcal{B}}}\Big[\overleftarrow{\mathcal{TC}}_m(x_{P_m})\Big]. \qquad (18)$$

Unlike its forward counterpart, reverse conditional total correlation evaluates the plausibility of samples drawn from an independent proposal and is sensitive to support mismatch.

**Comparison.** The distinction between the two directions mirrors the distinction between information loss and sampling risk. Conditional total correlation quantifies how much dependence is ignored on average under the target distribution, while reverse conditional total correlation quantifies how often and how severely independent sampling produces implausible configurations. As illustrated in Figure 2, distributions with strong but fully supported dependencies may exhibit large conditional total correlation but small reverse conditional total correlation, whereas near-zero-support configurations can induce very large reverse conditional total correlation despite small forward KL.

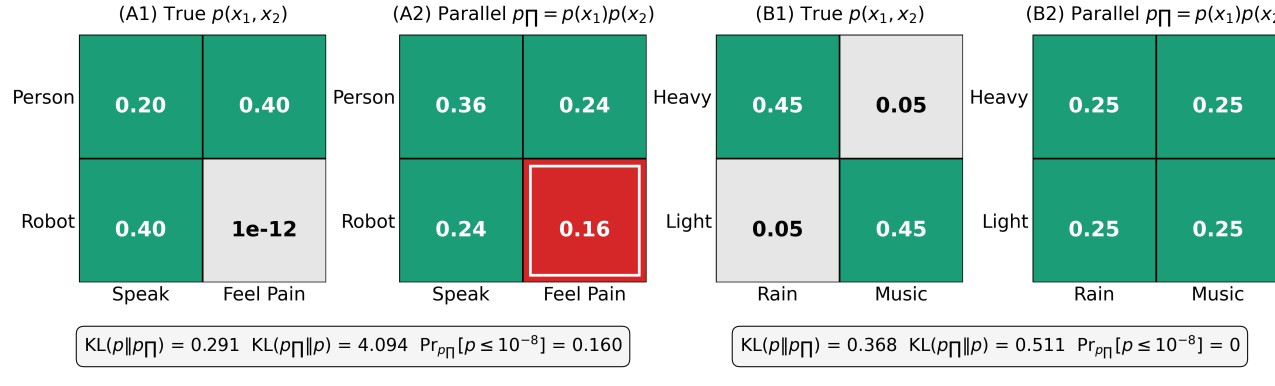

*Figure 2.* Forward vs. reverse KL under parallel factorization. (A1–A2) A distribution with a near-zero-support configuration. Although the factorized proposal $p_{\prod}(x_1, x_2) = p(x_1)p(x_2)$ preserves marginals, it assigns substantial probability mass to an implausible region (highlighted in red), resulting in small forward KL but large reverse KL and a non-negligible probability of incoherent samples. (B1–B2) A correlated but fully supported distribution. Here, factorization increases forward KL but does not introduce support violations, leading to small reverse KL and coherent parallel samples. All KL values are reported in nats; zero-probability entries are replaced by a small constant for numerical stability.

### 5.3. Verification and the Cost of Eliminating Sampling Error

The preceding analysis characterizes the source of sampling error in factorized parallel decoding. In principle, this error can be completely eliminated by exact verification, which produces samples from the target distribution $p$ and therefore yields zero forward and reverse KL divergence. However, the computational cost of achieving this correction is governed by a different quantity: the forward KL divergence between the true joint conditional distribution and the proposal used for parallel decoding.

**Proposition 5.1** (Verification Requires Exponential Cost).
*Let $p(x_{B_m} \mid x_{P_m})$ denote the true joint conditional distribution of a block, and let $p_{\prod}(x_{B_m} \mid x_{P_m})$ be the (factorized) blockwise proposal used by parallel decoding (e.g., $p_{\prod}(x_{B_m} \mid x_{P_m}) = \prod_{i \in B_m} p(x_i \mid x_{P_m})$ as in Eq. (5)). Any verification or accept–reject procedure that produces an exact sample from $p(x_{B_m} \mid x_{P_m})$ using proposals drawn from $p_{\prod}(x_{B_m} \mid x_{P_m})$ requires, in expectation, at least*

$$\mathbb{E}[\text{\# proposals}] \geq \exp\left(\overrightarrow{\mathcal{TC}}_m(x_{P_m})\right) \quad (19)$$

*proposals.*

**Interpretation.** This lower bound can be derived via standard rejection-sampling arguments, with an information-theoretic interpretation. We include a complete proof of Poposition 5.1 in App. C.2. When samples are drawn from a proposal distribution $p_{\prod}(x_{B_m} \mid x_{P_m})$ and accepted to exactly reproduce a target distribution $p(x_{B_m} \mid x_{P_m})$, the expected number of proposals required grows exponentially in the forward TC $\overrightarrow{\mathcal{TC}}_m(x_{P_m})$, which measures how difficult it is to correct the mismatch introduced by factorizing the joint conditional distribution.

> **Implication.** Verification can eliminate sampling error only by paying an exponential cost governed by the conditional TC. Practical parallel decoding methods avoid this cost and therefore necessarily retain structural sampling error.

### 5.4. Mitigating Sampling Error via Remasking

While exact verification is required to *eliminate* sampling error, many masked diffusion models employ iterative remasking as a practical heuristic to reduce incoherence. By repeatedly resampling subsets of variables, remasking can suppress locally inconsistent assignments and thereby reduce the *frequency* of visibly implausible samples.

However, iterative remasking does not fundamentally alter the class of proposal distributions used during decoding. As long as each remasking step relies on factorized or approximate conditionals that do not explicitly enforce joint support constraints, the resulting transition kernel may assign nonzero probability to configurations that are outside the support of the true conditional distribution. In this case, the reverse conditional divergence remains strictly positive and may be infinite.

Appendix C.5 formalizes this limitation by showing that, under a mild residual support violation assumption, no finite number of remasking steps can guarantee vanishing reverse KL divergence. Accordingly, remasking should be viewed as a heuristic for improving empirical sample quality rather than as a mechanism for ensuring distributional correctness.

> **Implication.** Iterative remasking can reduce incoherence in practice, but without explicit support containment or verification, it provides no guarantee of eliminating sampling error.

# 6. Experiments

## 6.1. Sampling Error under Parallel Decoding in a Block-HMM

We study sampling error induced by parallel factorization in a fully controlled generative model, where the target distribution is known exactly and model error is absent.

**Block-HMM with parity emissions.** We consider a binary sequence $X_{1:T} \in \{0,1\}^T$ partitioned into blocks $X^{(n)} \in \{0,1\}^B$. At the block level, generation is governed by a $K$-state hidden Markov model with latent states $Z_n$:

$$Z_1 \sim \pi, \qquad Z_n \mid Z_{n-1} \sim A. \tag{20}$$

Each block consists of one parity bit and $B-1$ content bits, $X^{(n)} = (P_n, Y_{n,1}, \ldots, Y_{n,B-1})$. Conditioned on $Z_n = z$, the content bits are independent Bernoulli variables,

$$Y_{n,i} \mid Z_n = z \sim \mathrm{Bern}(\rho_z), \tag{21}$$

where $\{\rho_z\}_{z=1}^K$ are fixed, state-specific parameters shared across all blocks. The parity bit then satisfies an XOR constraint with noise

$$P_n = \bigoplus_{i=1}^{B-1} Y_{n,i}, \qquad \Pr(P_n \neq \oplus_i Y_{n,i}) = \eta. \tag{22}$$

As $\eta \to 0$, the model induces strong intra-block dependence and near-zero-support configurations.

Detailed parameter configuration is shown in App. E.

**Parallel decoding.** We compare two decoding procedures. Mean-field parallel decoding samples each block independently from exact marginal conditionals,

$$p_{\prod}(x^{(n)} \mid x_{<n}) = \prod_{i \in B} p(x_i \mid x_{<n}), \tag{23}$$

which preserves all marginals but discards intra-block dependencies. As a control, verified decoding samples blocks exactly from $p(x^{(n)} \mid x_{<n})$ by enumeration over all $2^B$ block configurations, producing samples from the true joint distribution.

**Evaluation metrics.** We evaluate decoding quality using three complementary metrics. Sampling risk is measured by the reverse KL divergence $\mathrm{KL}(p_{\prod}\|p)$, forward KL divergence $\mathrm{KL}(p\|p_{\prod})$ and the incoherence rate, where incoherence rate is reported as the fraction of blocks whose true conditional probability falls below the threshold, averaged over blocks and Monte Carlo samples, i.e.,

$$\Pr_{x \sim q}\Big[ p(x^{(n)} \mid x_{<n}) \leq \tau \Big], \tag{24}$$

In the implementation we set $\tau = 10^{-8}$.

*Table 1.* **Unmasking Strategy Comparison in LLaDA (10-digit Addition).** Overall accuracy (%) with edge case accuracy in parentheses.

| Strategy | 16 | 32 | 64 | 128 |
|---|---|---|---|---|
| Left-to-Right | 76.0 (45.0) | 69.0 (50.0) | 79.0 (50.0) | 84.0 (62.5) |
| Random | 87.0 (80.0) | 75.0 (77.5) | 78.0 (75.0) | 79.0 (80.0) |
| Right-to-Left | 93.0 (82.5) | 87.0 (85.0) | 94.0 (85.0) | 92.0 (80.0) |
| **Low-Confidence** | **97.0** (95.0) | **89.0** (80.0) | **99.0** (97.5) | **99.0** (97.5) |

**Results.** Figure 3 reports results for block size $B = 8$ across parity noise levels $\eta$. Mean-field decoding incurs extremely large reverse KL in the low-noise regime, exceeding 70 nats when $\eta = 10^{-8}$, while verified decoding achieves zero reverse KL. In contrast, the forward KL remains relatively stable ($\approx$ 11–13 nats) across all noise levels, indicating that it is insensitive to near-zero-support configurations. The incoherence rate closely tracks reverse KL: when $\eta$ is small, mean-field decoding produces incoherent blocks with probability close to $50\%$, corresponding to frequent violations of the parity constraint.

## 6.2. Empirical Validation: Arithmetic Reasoning

To empirically validate the "Easy-first" principle, we evaluated autoregressive models on a **10-digit addition** task (see App. D for detailed experimental setup). In this task, the logical flow of information—governed by carry propagation—moves from the least significant digit (LSD) to the most significant digit (MSD). Consequently, a **Right-to-Left (R2L)** generation sequence aligns with the optimal "Easy-First" order by resolving low-entropy, local dependencies before global ones. Conversely, a standard **Left-to-Right (L2R)** approach forces the model to predict high-entropy tokens (MSDs) without the necessary computational context provided by the lower digits.

The results, illustrated in Figure 4, demonstrate a clear relationship between model capacity and order sensitivity: when a model is near-perfect, generation order is nearly invariant; however, as approximation errors increase, the "Easy-First" (R2L) path becomes essential for suppressing rollout-induced covariate shift.

We further test this task in the parallel decoding regime using LLaDA-8B-Instruct (Nie et al., 2025) on addition tasks. We compare four unmasking strategies: **Low-Confidence** (prioritizing high-confidence tokens), **R2L**, **L2R**, and **Random**. Results in Table 1 show that the **Low-Confidence** strategy achieves the highest accuracy (up to 99%), particularly excelling in edge cases. While R2L remains a strong heuristic, it is surpassed by the confidence-based approach. This confirms that the "Easy-First" principle is fundamentally about resolving *low-entropy variables first*; in parallel decoding, model confidence serves as a superior proxy for easiness compared to fixed spatial orders.

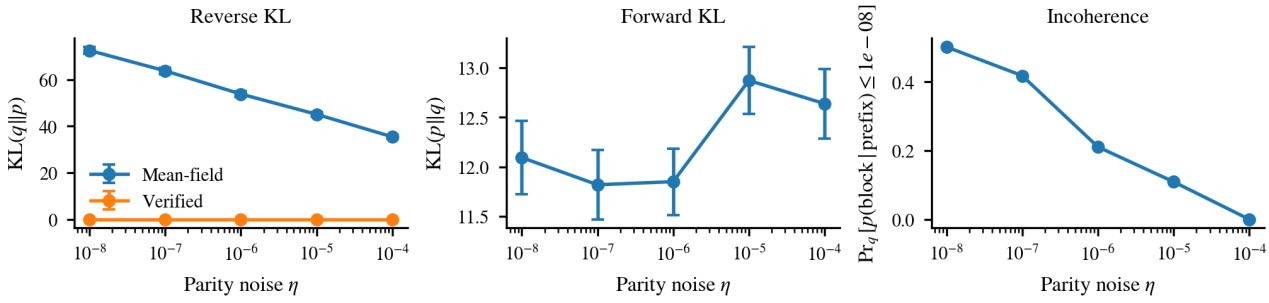

*Figure 3.* Forward KL, reverse KL, and incoherence under parallel mean-field decoding in a Block-HMM with parity emissions ($B = 8$). Mean-field decoding incurs severe sampling error and incoherence at low parity noise despite modest forward KL, while verified decoding achieves zero sampling error.

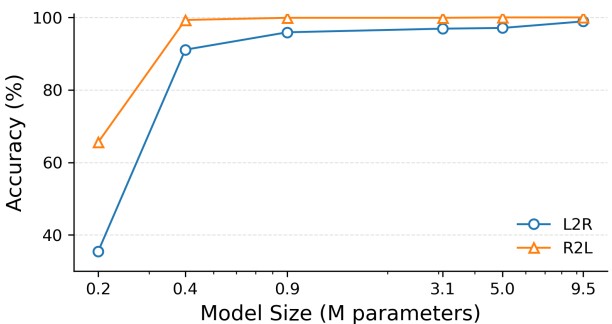

*Figure 4.* Comparison of L2R and R2L decoding strategies under varying model scales.

## 7. Related Work

**Masked Diffusion and Discrete Flow Matching.** Diffusion models have recently been adapted from continuous domains (Ho et al., 2020; Song et al., 2020) to discrete sequence generation. Early work such as D3PM and MaskGIT introduced discrete corruption and iterative refinement (Austin et al., 2021; Chang et al., 2022). More recent methods scale these ideas to language modeling, including MDLM (Sahoo et al., 2024) and LLaDA (Nie et al., 2025), as well as Discrete Flow Matching, which provides an optimal-transport interpretation of discrete generation paths (Campbell et al., 2024; Gat et al., 2024). These works primarily focus on training objectives and architectures. In contrast, we study *inference-time dynamics*, showing that the generation order—equivalently, the integration path—fundamentally determines error accumulation.

**Parallel and Speculative Decoding.** To mitigate the $O(T)$ decoding latency of autoregressive models, speculative decoding adopts a draft-and-verify paradigm that preserves the target distribution under exact verification (Leviathan et al., 2023; Chen et al., 2023). Pure parallel decoding methods, including blockwise decoding (Stern et al., 2018) and aggressive unmasking in MDMs, relax verification to

maximize throughput.

**Generation Order and Easy-First Strategies.** The importance of generation order has long been recognized in structured prediction and non-autoregressive translation (Gu et al., 2017). Recent work exploits model uncertainty to guide generation trajectories, including confidence-based masking (Chang et al., 2022; Sahoo et al., 2024) and path- or edit-based planning (Peng et al., 2025; Havasi et al., 2025). While these strategies utilize uncertainty as a heuristic, the field has lacked a unified theoretical framework to separate the distinct failure modes of order sensitivity and parallelization bias.

Furthermore, Chen et al. (2025) provide theoretical analysis of block-size schedules under mean-field parallel decoding, where the variables within each block are randomly selected. Their analysis highlights the role of the conditional total correlation in minimizing the forward KL error (Eq. (16)), averaged over block selections.

## 8. Conclusion

We investigated generation order and parallel decoding in masked diffusion models through an information-theoretic lens. We showed that under model error, generation order is intrinsically important, with Easy-First strategies becoming more effective as error increases. We further demonstrated that factorized parallel decoding introduces unavoidable sampling bias, which can lead to severe incoherence that forward KL fails to capture. While exact verification removes this bias at exponential cost, practical heuristics such as remasking cannot guarantee distributional correctness.

Overall, our results clarify the fundamental trade-off between decoding efficiency and fidelity, and provide principled guidance for parallel generation. Future avenues include analyzing the **coupled dynamics** of order and parallelism, and developing decoding schemes that jointly reason about error propagation and dependency structure.

## Impact Statement

This paper presents a theoretical analysis of decoding algorithms for generative models. Our findings clarify how generation order and parallelization affect distributional correctness, with implications for efficient and reliable inference. As this work focuses on decoding behavior rather than new modeling capabilities, it does not introduce societal risks beyond those already associated with existing generative models.

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

## A. Proofs for Section 4

### A.1. Proof of the Order-Dependent KL Decomposition

We prove Eq. 7.

Let $\pi$ be an arbitrary permutation of $\{1, \ldots, T\}$ and define the model-induced distribution

$$p_\theta^\pi(x_{1:T}) = \prod_{t=1}^{T} p_\theta(x_{\pi(t)} \mid x_{\pi(<t)}). \tag{25}$$

Similarly, the true data distribution factorizes as

$$p_{\text{data}}(x_{1:T}) = \prod_{t=1}^{T} p_{\text{data}}(x_{\pi(t)} \mid x_{\pi(<t)}), \tag{26}$$

which holds for any permutation by the chain rule of probability.

By definition, the reverse KL divergence is

$$\text{KL}(p_\theta^\pi \,\|\, p_{\text{data}}) = \mathbb{E}_{x \sim p_\theta^\pi} \left[ \log \frac{p_\theta^\pi(x)}{p_{\text{data}}(x)} \right]. \tag{27}$$

Substituting the factorizations yields

$$\log \frac{p_\theta^\pi(x)}{p_{\text{data}}(x)} = \sum_{t=1}^{T} \log \frac{p_\theta(x_{\pi(t)} \mid x_{\pi(<t)})}{p_{\text{data}}(x_{\pi(t)} \mid x_{\pi(<t)})}. \tag{28}$$

Taking expectation under $x \sim p_\theta^\pi$ and exchanging summation with expectation gives

$$\text{KL}(p_\theta^\pi \,\|\, p_{\text{data}}) = \sum_{t=1}^{T} \mathbb{E}_{x \sim p_\theta^\pi} \left[ \log \frac{p_\theta(x_{\pi(t)} \mid x_{\pi(<t)})}{p_{\text{data}}(x_{\pi(t)} \mid x_{\pi(<t)})} \right]. \tag{29}$$

Applying iterated expectation with respect to the prefix distribution $x_{\pi(<t)} \sim p_\theta^\pi$ yields

$$\text{KL}(p_\theta^\pi \,\|\, p_{\text{data}}) = \sum_{t=1}^{T} \mathbb{E}_{x_{\pi(<t)} \sim p_\theta^\pi} \left[ \text{KL}\big(p_\theta(\cdot \mid x_{\pi(<t)}) \,\|\, p_{\text{data}}(\cdot \mid x_{\pi(<t)})\big) \right]. \tag{30}$$

Defining

$$\epsilon_\theta(x_{\pi(<t)}) = \text{KL}\big(p_\theta(\cdot \mid x_{\pi(<t)}) \,\|\, p_{\text{data}}(\cdot \mid x_{\pi(<t)})\big) \tag{31}$$

establishes Eq. 7.

### A.2. Derivation of the Weighted Error Objective

*Proof of Eq.* (11). Define $e_t := \bar{\epsilon}_{\pi(t)}$ and $h_t := H\big(X_{\pi(t)} \mid x_{\pi(<t)}\big)$. The cumulative objective is

$$\mathcal{J}(\pi) := \sum_{t=1}^{T} e_t. \tag{32}$$

By Assumption 4.2, for each $t$ we have

$$e_t \leq \alpha h_t + \lambda \sum_{k=1}^{t-1} e_k + b. \tag{33}$$

Substituting (33) into the definition of $\mathcal{J}(\pi)$ yields

$$\mathcal{J}(\pi) \leq \sum_{t=1}^{T} \alpha h_t + \lambda \sum_{t=1}^{T} \sum_{k=1}^{t-1} e_k + Tb. \tag{34}$$

Reordering the double sum gives

$$\sum_{t=1}^{T}\sum_{k=1}^{t-1} e_k = \sum_{k=1}^{T-1}(T-k)\,e_k. \tag{35}$$

We now unroll the recursion. From (33), the entropy term $h_t$ contributes $\alpha h_t$ directly to $e_t$, and propagates to later errors multiplicatively via the prefix term. Specifically, its contribution to $e_{t+s}$ is bounded by $\alpha\lambda^s h_t$ for $s \geq 0$. Summing over all future steps yields the total weight

$$W_t = \alpha \sum_{s=0}^{T-t} \lambda^s = \alpha(1+\lambda)^{T-t}. \tag{36}$$

Collecting all entropy terms, we obtain

$$\mathcal{J}(\pi) \leq \sum_{t=1}^{T} W_t\, H\big(X_{\pi(t)} \mid x_{\pi(<t)}\big) + \tilde{b}, \tag{37}$$

where $\tilde{b} := Tb$ is independent of the generation order $\pi$. $\qquad\square$

### A.3. Consequence for Ordering

Because $W_t$ decays exponentially in $t$, uncertainty incurred at early steps dominates the total objective. Therefore, any ordering that delays low-entropy variables to later positions incurs a strictly larger objective under the same entropy trajectory. This structural asymmetry motivates greedy, easy-first approximations that minimize immediate conditional entropy.

## B. Empirical Validation of Assumption 4.2

To validate our core theoretical assumption, we conduct an empirical study measuring the relationship between teacher model entropy and student model error during autoregressive generation.

**Experimental Setup.** We use Qwen2.5-7B as the teacher model and Qwen2.5-0.5B as the student model. Starting from 100 diverse seed prompts, we generate sequences of varying lengths (20, 100, 200 tokens) using greedy decoding from the student model. At each step $t$, we measure: (1) the teacher's conditional entropy $H_t = H(X_t \mid x_{<t})$, and (2) the KL divergence $\epsilon_t = \mathrm{KL}(p_{\text{teacher}}(\cdot \mid x_{<t}) \| p_{\text{student}}(\cdot \mid x_{<t}))$ between teacher and student distributions. We then fit the linear model:

$$\epsilon_t = \alpha \cdot H_t + \lambda \cdot \bar{\epsilon}_{<t} + \eta_t, \tag{38}$$

where $\bar{\epsilon}_{<t} = \frac{1}{t}\sum_{k<t}\epsilon_k$ is the average past error.

**Results.** Figure 5 shows results for sequences of 200 tokens. The entropy coefficient $\alpha = 0.138$ is highly significant ($p < 0.001$, $R^2 = 0.26$), confirming that higher teacher uncertainty correlates with larger student errors.

These results provide strong empirical support for Assumption 4.2: the local error at each generation step is primarily driven by intrinsic uncertainty (entropy), with error propagation playing a secondary role.

## C. Proofs for Section 5

### C.1. Forward- and Reverse-KL Blockwise Decompositions

We restate the blockwise decoding distribution

$$p_{\mathcal{B}}(x_{1:T}) = \prod_{m=1}^{M} p_{\Pi}(x_{B_m} \mid x_{P_m}), \tag{39}$$

and the target joint distribution factorized by the chain rule over the same blocks,

$$p(x_{1:T}) = \prod_{m=1}^{M} p(x_{B_m} \mid x_{P_m}). \tag{40}$$

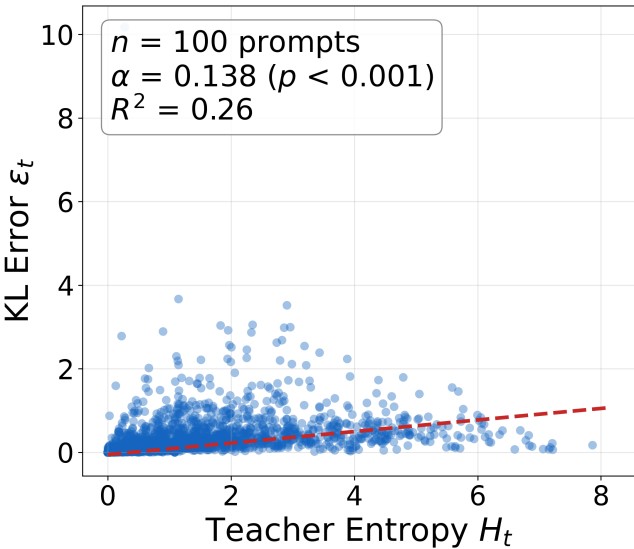

*Figure 5.* Teacher entropy $H_t$ vs. KL error $\epsilon_t$ (200 tokens). Higher entropy correlates with larger student error ($\alpha = 0.138$, $p < 0.001$, $R^2 = 0.26$).

**Forward KL decomposition.** Using the definition of KL divergence and Eq. (39)–Eq. (40),

$$\mathrm{KL}(p \,\|\, p_{\mathcal{B}}) = \mathbb{E}_{x \sim p}\left[\log \frac{p(x_{1:T})}{p_{\mathcal{B}}(x_{1:T})}\right] = \mathbb{E}_{x \sim p}\left[\sum_{m=1}^{M} \log \frac{p(x_{B_m} \mid x_{P_m})}{p_{\prod}(x_{B_m} \mid x_{P_m})}\right]. \tag{41}$$

Swap sum and expectation and condition on $x_{P_m}$ under $p$:

$$\mathrm{KL}(p \,\|\, p_{\mathcal{B}}) = \sum_{m=1}^{M} \mathbb{E}_{x_{P_m} \sim p}\left[\mathbb{E}_{x_{B_m} \sim p(\cdot \mid x_{P_m})}\left[\log \frac{p(x_{B_m} \mid x_{P_m})}{p_{\prod}(x_{B_m} \mid x_{P_m})}\right]\right] = \sum_{m=1}^{M} \mathbb{E}_{x_{P_m} \sim p}\left[\mathrm{KL}\big(p(\cdot \mid x_{P_m}) \,\|\, p_{\prod}(\cdot \mid x_{P_m})\big)\right]. \tag{42}$$

If the proposal factorizes within the block as $p_{\prod}(x_{B_m} \mid x_{P_m}) = \prod_{i \in B_m} p(x_i \mid x_{P_m})$, then each inner term becomes

$$\mathrm{KL}\left(p(x_{B_m} \mid x_{P_m}) \,\Big\|\, \prod_{i \in B_m} p(x_i \mid x_{P_m})\right), \tag{43}$$

i.e., the (forward) conditional total correlation in block $B_m$ given $x_{P_m}$.

**Reverse KL decomposition.** Similarly,

$$\mathrm{KL}(p_{\mathcal{B}} \,\|\, p) = \mathbb{E}_{x \sim p_{\mathcal{B}}}\left[\log \frac{p_{\mathcal{B}}(x_{1:T})}{p(x_{1:T})}\right] = \mathbb{E}_{x \sim p_{\mathcal{B}}}\left[\sum_{m=1}^{M} \log \frac{p_{\prod}(x_{B_m} \mid x_{P_m})}{p(x_{B_m} \mid x_{P_m})}\right]. \tag{44}$$

Again swapping sum and expectation and conditioning on $x_{P_m}$ under $p_{\mathcal{B}}$ yields

$$\mathrm{KL}(p_{\mathcal{B}} \,\|\, p) = \sum_{m=1}^{M} \mathbb{E}_{x_{P_m} \sim p_{\mathcal{B}}}\left[\mathrm{KL}\big(p_{\prod}(\cdot \mid x_{P_m}) \,\|\, p(\cdot \mid x_{P_m})\big)\right]. \tag{45}$$

Under the factorized proposal $p_{\prod}(x_{B_m} \mid x_{P_m}) = \prod_{i \in B_m} p(x_i \mid x_{P_m})$, the inner KL becomes

$$\mathrm{KL}\left(\prod_{i \in B_m} p(x_i \mid x_{P_m}) \,\Big\|\, p(x_{B_m} \mid x_{P_m})\right), \tag{46}$$

which is the reverse conditional total correlation used in Section 5.2. $\qquad\square$

## C.2. Proof of Proposition 5.1

Fix a block index $m$ and a realized context $x_{P_m}$. Let the target conditional be $p(\cdot \mid x_{P_m})$ over $x_{B_m}$ and the proposal be $p_\prod(\cdot \mid x_{P_m})$. Consider an accept–reject (rejection sampling) procedure that draws i.i.d. proposals $X \sim p_\prod(\cdot \mid x_{P_m})$ and accepts each draw with probability $a(X) \in [0, 1]$ such that the distribution of accepted samples is exactly $p(\cdot \mid x_{P_m})$.

For correctness, the accepted-sample density must satisfy, for all $x_{B_m}$,

$$p(x_{B_m} \mid x_{P_m}) = \frac{p_\prod(x_{B_m} \mid x_{P_m}) \, a(x_{B_m})}{Z}, \tag{47}$$

where $Z := \mathbb{E}_{X \sim p_\prod(\cdot \mid x_{P_m})}[a(X)]$ is the overall acceptance probability. Rearranging Eq. (47) gives

$$a(x_{B_m}) = Z \, \frac{p(x_{B_m} \mid x_{P_m})}{p_\prod(x_{B_m} \mid x_{P_m})}. \tag{48}$$

Since $a(x_{B_m}) \leq 1$ for all $x_{B_m}$, we must have

$$Z \leq \inf_{x_{B_m}} \frac{p_\prod(x_{B_m} \mid x_{P_m})}{p(x_{B_m} \mid x_{P_m})}. \tag{49}$$

Equivalently, defining the usual rejection constant

$$M(x_{P_m}) := \sup_{x_{B_m}} \frac{p(x_{B_m} \mid x_{P_m})}{p_\prod(x_{B_m} \mid x_{P_m})}, \tag{50}$$

we obtain

$$Z \leq \frac{1}{M(x_{P_m})}. \tag{51}$$

Because each proposal is accepted with probability $Z$, the number of proposals $N$ until the first acceptance is geometric with mean

$$\mathbb{E}[N \mid x_{P_m}] = \frac{1}{Z} \geq M(x_{P_m}). \tag{52}$$

It remains to lower bound $M(x_{P_m})$ by an exponential in the forward KL. From Eq. (50), for every $x_{B_m}$,

$$\log M(x_{P_m}) \geq \log \frac{p(x_{B_m} \mid x_{P_m})}{p_\prod(x_{B_m} \mid x_{P_m})}. \tag{53}$$

Taking expectation over $x_{B_m} \sim p(\cdot \mid x_{P_m})$ yields

$$\log M(x_{P_m}) \geq \mathbb{E}_{x_{B_m} \sim p(\cdot \mid x_{P_m})} \left[ \log \frac{p(x_{B_m} \mid x_{P_m})}{p_\prod(x_{B_m} \mid x_{P_m})} \right] = \mathrm{KL}\big(p(\cdot \mid x_{P_m}) \,\|\, p_\prod(\cdot \mid x_{P_m})\big). \tag{54}$$

Exponentiating Eq. (54) gives

$$M(x_{P_m}) \geq \exp\big(\mathrm{KL}\big(p(\cdot \mid x_{P_m}) \,\|\, p_\prod(\cdot \mid x_{P_m})\big)\big). \tag{55}$$

Combining Eq. (52) and Eq. (55), we conclude

$$\mathbb{E}[N \mid x_{P_m}] \geq \exp\big(\mathrm{KL}\big(p(\cdot \mid x_{P_m}) \,\|\, p_\prod(\cdot \mid x_{P_m})\big)\big). \tag{56}$$

Finally, under the factorized parallel proposal in Eq. (5),

$$p_\prod(x_{B_m} \mid x_{P_m}) = \prod_{i \in B_m} p(x_i \mid x_{P_m}), \tag{57}$$

the KL in Eq. (56) is exactly the forward conditional total correlation $\overrightarrow{\mathcal{TC}}_m(x_{P_m})$ defined in Section 5.2. Therefore,

$$\mathbb{E}[\text{\# proposals} \mid x_{P_m}] \geq \exp\left(\overrightarrow{\mathcal{TC}}_m(x_{P_m})\right), \tag{58}$$

which is Proposition 5.1.

### C.3. Support Mismatch Implies Unbounded Reverse KL

This subsection formalizes the claim that reverse-KL-type sampling risk can blow up under support mismatch, even when marginals are preserved.

**Lemma C.1** (Reverse KL diverges under conditional support mismatch). *Fix a block $B_m$ and context $x_{P_m}$. If there exists an assignment $\tilde{x}_{B_m}$ such that*

$$p_{\prod}(\tilde{x}_{B_m} \mid x_{P_m}) > 0 \qquad \text{and} \qquad p(\tilde{x}_{B_m} \mid x_{P_m}) = 0, \tag{59}$$

*then*

$$\mathrm{KL}\big(p_{\prod}(\cdot \mid x_{P_m}) \,\|\, p(\cdot \mid x_{P_m})\big) = +\infty. \tag{60}$$

*Consequently, if $\Pr_{x_{P_m} \sim p_{\mathcal{B}}}[\, \text{Eq. (59) holds} \,] > 0$, then*

$$\mathrm{KL}(p_{\mathcal{B}} \,\|\, p) = +\infty. \tag{61}$$

*Proof.* By definition,

$$\mathrm{KL}\big(p_{\prod}(\cdot \mid x_{P_m}) \,\|\, p(\cdot \mid x_{P_m})\big) = \sum_{x_{B_m}} p_{\prod}(x_{B_m} \mid x_{P_m}) \log \frac{p_{\prod}(x_{B_m} \mid x_{P_m})}{p(x_{B_m} \mid x_{P_m})}. \tag{62}$$

If Eq. (59) holds, then the summand at $x_{B_m} = \tilde{x}_{B_m}$ equals

$$p_{\prod}(\tilde{x}_{B_m} \mid x_{P_m}) \log \frac{q(\tilde{x}_{B_m} \mid x_{P_m})}{0} = p_{\prod}(\tilde{x}_{B_m} \mid x_{P_m}) \cdot (+\infty) = +\infty, \tag{63}$$

which implies Eq. (60).

For the global statement, use the blockwise reverse-KL decomposition from Eq. (45):

$$\mathrm{KL}(p_{\mathcal{B}} \,\|\, p) = \sum_{m=1}^{M} \mathbb{E}_{x_{P_m} \sim p_{\mathcal{B}}} \big[ \mathrm{KL}\big(p_{\prod}(\cdot \mid x_{P_m}) \,\|\, p(\cdot \mid x_{P_m})\big) \big]. \tag{64}$$

If with positive probability over $x_{P_m} \sim p_{\mathcal{B}}$ the inner KL is $+\infty$, then the expectation is $+\infty$ for that $m$, hence the sum is $+\infty$. □

**Corollary C.2** (Reverse conditional total correlation can be unbounded). *Under the factorized proposal $p_{\prod}(x_{B_m} \mid x_{P_m}) = \prod_{i \in B_m} p(x_i \mid x_{P_m})$, if there exists $\tilde{x}_{B_m}$ satisfying Eq. (59), then the reverse conditional total correlation*

$$\overleftarrow{\mathcal{TC}}_m(x_{P_m}) = \mathrm{KL}\Bigg( \prod_{i \in B_m} p(x_i \mid x_{P_m}) \,\Bigg\|\, p(x_{B_m} \mid x_{P_m}) \Bigg) \tag{65}$$

*is infinite.*

### C.4. Reverse KL vs. Incoherence Probability

This subsection relates event-level "incoherence" to reverse KL in a simple way.

**Lemma C.3** (A one-sided bound from reverse KL). *Let $A_\epsilon := \{x : p(x) \leq \epsilon\}$ for $\epsilon \in (0, 1]$. Then for any decoding distribution $p_{\mathcal{B}}$,*

$$\Pr_{x \sim p_{\mathcal{B}}}\big[x \in A_\epsilon\big] \leq \frac{\mathbb{E}_{x \sim p_{\mathcal{B}}}\big[-\log p(x)\big]}{\log(1/\epsilon)}. \tag{66}$$

*Proof.* Define the nonnegative random variable $Z := -\log p(X)$ for $X \sim p_{\mathcal{B}}$. Then $X \in A_\epsilon$ implies $p(X) \leq \epsilon$, i.e.,

$$-\log p(X) \geq \log(1/\epsilon). \tag{67}$$

By Markov's inequality,

$$\Pr[Z \geq \log(1/\epsilon)] \leq \frac{\mathbb{E}[Z]}{\log(1/\epsilon)}, \tag{68}$$

which is exactly Eq. (66). □

**Remark.** Since

$$\mathbb{E}_{x \sim p_{\mathcal{B}}}\big[-\log p(x)\big] = \mathrm{KL}(p_{\mathcal{B}} \| p) + \mathbb{H}(p_{\mathcal{B}}), \tag{69}$$

Lemma C.3 shows that small reverse KL *together with* controlled sampler entropy implies a small incoherence probability for any fixed threshold $\epsilon$.

### C.5. Limitations of Iterative Remasking Without Verification

In this appendix, we formalize the claim made in Sec. 5.4 that iterative remasking, while empirically effective at reducing incoherence, cannot in general eliminate the sampling error induced by factorized parallel decoding in the absence of explicit verification or support containment.

#### C.5.1. SETUP

Let $p(x_B \mid x_P)$ denote the true conditional distribution of a block of variables $X_B$ given context $X_P$. We assume that $p$ exhibits strict support constraints, i.e., there exist configurations $x_B$ such that

$$p(x_B \mid x_P) = 0. \tag{70}$$

We consider an iterative remasking procedure defined by a Markov transition kernel $K$ over $X_B$. Each remasking iteration samples a new configuration conditioned on the current context (and possibly a partially masked previous configuration). We assume that proposals within each iteration are generated by a factorized or approximate conditional mechanism, without accept–reject steps or other explicit verification procedures.

#### C.5.2. RESIDUAL SUPPORT VIOLATION ASSUMPTION

We make the following minimal assumption.

**Assumption C.4. (Residual Support Violation).** There exist a context $x_P$ and a configuration $x_B$ such that

$$p(x_B \mid x_P) = 0 \quad \text{and} \quad K(x_B \mid x_P) > 0. \tag{71}$$

That is, a single remasking step assigns nonzero probability to at least one configuration outside the support of the true conditional distribution.

This assumption merely states that remasking does not perfectly enforce joint support constraints.

#### C.5.3. MAIN RESULT

**Proposition C.5** (Remasking Cannot Guarantee Elimination of Sampling Error). *Let $q^{(n)}(\cdot \mid x_P)$ denote the conditional distribution over $X_B$ induced after $n$ iterations of remasking, starting from any initial distribution $q^{(0)}$. Under Assumption C.4, for any finite $n \geq 1$,*

$$\mathrm{KL}\Big(q^{(n)}(\cdot \mid x_P) \, \| \, p(\cdot \mid x_P)\Big) = +\infty, \tag{72}$$

*or equivalently, the reverse KL divergence cannot be guaranteed to vanish.*

*Proof.* By Assumption C.4, there exists a configuration $x_B$ such that $p(x_B \mid x_P) = 0$ and $K(x_B \mid x_P) > 0$. For any initial distribution $q^{(0)}$, the distribution after one remasking step satisfies

$$q^{(1)}(x_B \mid x_P) = \sum_{x'_B} q^{(0)}(x'_B \mid x_P) \, K(x_B \mid x'_B, x_P) \tag{73}$$

$$\geq K(x_B \mid x_P) > 0, \tag{74}$$

where the inequality follows from marginalizing over all previous states.

Thus, $q^{(1)}$ assigns strictly positive probability mass to a configuration $x_B$ that lies outside the support of $p(\cdot \mid x_P)$. By Lemma B.3 (support mismatch implies infinite reverse KL), this implies

$$\mathrm{KL}\Big(q^{(1)}(\cdot \mid x_P) \, \| \, p(\cdot \mid x_P)\Big) = +\infty. \tag{75}$$

The same argument applies to any finite number of iterations $n$, since the remasking kernel continues to assign nonzero probability to support-violating configurations. Therefore, iterative remasking alone cannot guarantee elimination of sampling error under reverse KL divergence. $\square$

### C.5.4. DISCUSSION

Proposition C.5 does not preclude iterative remasking from improving sample quality in practice. By repeatedly resampling under updated contexts, remasking may reduce the *frequency* of incoherent configurations and concentrate probability mass on higher-likelihood regions. However, without an explicit mechanism enforcing support containment—such as accept–reject verification—remasking provides no worst-case guarantee of distributional correctness.

## D. Experimental Details for Arithmetic Tasks

This section provides detailed experimental settings for the arithmetic reasoning experiments reported in Sec. 6.2, with a focus on the 10-digit addition task. We describe the dataset construction, difficulty stratification, model configurations, training procedure, and evaluation protocol.

### D.1. Dataset Construction and Difficulty Levels

To enable fine-grained analysis of generation order effects, we construct a synthetic test suite of 1,000 addition problems with controlled difficulty. All operands consist of non-negative integers with up to 10 digits. Test cases are stratified by operand length and carry structure.

**10-Digit Addition.** The test set is divided into six categories, following the exact generation procedure used during evaluation:

- **Easy (10%):** Addition of 1–3 digit numbers (e.g., $12 + 5$), requiring little or no carry propagation.

- **Medium (20%):** Addition of 4–6 digit numbers.

- **Hard (30%):** Addition of 7–9 digit numbers, involving longer carry chains.

- **Extreme (20%):** Full 10-digit addition (e.g., $1,234,567,890 + 9,876,543,210$), representing the most demanding cases in terms of long-range dependency.

- **Edge Cases (10%):** Pathological cases designed to stress-test carry propagation and boundary behavior, including:
    - *Cascading carries:* $9,999,999,999 + 1$.
    - *Zero handling:* Operations involving zero (e.g., $A + 0$).
    - *Near-maximum values:* Large operands close to the 10-digit limit.

- **Mixed (10%):** Operands with mismatched digit lengths (e.g., 10-digit + 2-digit).

All test cases are generated once using a fixed random seed and reused across all models to ensure fair comparison.

### D.2. Model Architecture and Training Configuration

We train Transformer-based autoregressive models (*NanoAdder*) from scratch. Each model operates at the character level with a vocabulary consisting of digits `0`--`9` and special symbols `+`, `=`, and padding.

**Architecture.** All models share the same architectural template, varying only in scale for the parameter sweep experiments:

- Context length: 48 tokens

- Learned token and positional embeddings

- Causal self-attention with a standard Transformer encoder stack

**Training Procedure.** Models are trained using next-token prediction with a masked loss: tokens before the = symbol are excluded from the loss to ensure the model is trained only to predict the result of the addition. Training examples are generated on-the-fly using a curriculum strategy where operand length is uniformly sampled up to 10 digits.

Optimization is performed using AdamW with cosine learning-rate decay and linear warmup. Gradient clipping is applied to stabilize training. Unless otherwise specified, all models are trained for 50k steps with a batch size of 256.

### D.3. Evaluation Protocol

Evaluation is performed using exact-match accuracy on the full predicted sum. A prediction is counted as correct if and only if the entire generated result matches the ground-truth integer exactly.

For Right-to-Left (R2L) models, the generated output string is reversed before comparison with the ground truth. All reported accuracies correspond to overall exact-match accuracy, as well as per-category accuracy aggregated over the predefined difficulty levels.

This evaluation protocol ensures that differences in performance reflect genuine reasoning errors rather than partial correctness or formatting artifacts.

## E. Experimental Details for the Block-HMM Study

This appendix provides implementation-level details for the Block-HMM experiments in Section 6.1.

### E.1. Model specification

We study sampling error under parallel decoding in a fully specified generative model, where the target distribution is known exactly and no model approximation error is present. The data distribution is defined by a Block-HMM with parity emissions.

**Latent dynamics.** Let $Z_n \in \{1, \ldots, K\}$ denote the latent state associated with block $n$. The latent states follow a first-order Markov chain

$$Z_1 \sim \pi, \qquad Z_n \mid Z_{n-1} \sim A. \tag{76}$$

We use a symmetric transition matrix parameterized by a single self-transition probability $a$:

$$A_{zz} = a, \qquad A_{zz'} = \frac{1-a}{K-1} \quad (z' \neq z). \tag{77}$$

Unless otherwise stated, we use $K = 2$, $\pi = (0.5, 0.5)$, and $a = 0.9$, yielding persistent latent modes across blocks.

### E.2. Block emission distribution

Each block $X^{(n)} \in \{0, 1\}^B$ consists of one parity bit and $B - 1$ content bits,

$$X^{(n)} = (P_n, Y_{n,1}, \ldots, Y_{n,B-1}). \tag{78}$$

Conditioned on the latent state $Z_n = z$, the content bits are independent Bernoulli variables,

$$Y_{n,i} \mid Z_n = z \sim \text{Bern}(\rho_z), \qquad i = 1, \ldots, B-1, \tag{79}$$

where $\{\rho_z\}_{z=1}^K$ are fixed, state-specific parameters shared across all blocks. In our experiments, we use

$$\rho = (0.9, 0.1).$$

The parity bit enforces a global XOR constraint with noise,

$$P_n = \bigoplus_{i=1}^{B-1} Y_{n,i}, \qquad \Pr(P_n \neq \oplus_i Y_{n,i}) = \eta. \tag{80}$$

As $\eta \to 0$, the model induces strong intra-block dependence and near-zero-support configurations.

### E.3. Exact conditional distributions

Given a prefix $x^{<n}$ ending at a block boundary, the true conditional distribution of the next block is

$$p(x^{(n)} \mid x^{<n}) = \sum_z p(z_n = z \mid x^{<n}) \, p(x^{(n)} \mid Z_n = z). \tag{81}$$

The posterior $p(z_n \mid x^{<n})$ is computed exactly using standard HMM forward filtering. For a fixed block size $B$, the conditional distribution $p(x^{(n)} \mid x^{<n})$ is obtained by explicit enumeration over all $2^B$ possible block configurations.

### E.4. Decoding procedures

We compare two decoding strategies.

**Mean-field parallel decoding.** At each block boundary, we compute the exact marginal probabilities

$$p(x_{n,i} = 1 \mid x^{<n}), \quad i = 1, \ldots, B,$$

and sample each bit independently according to these marginals. This preserves all marginal conditionals but discards intra-block dependencies.

**Verified decoding (oracle baseline).** As a control, we use a verified decoding procedure that samples each block exactly from the true conditional distribution $p(x^{(n)} \mid x^{<n})$. This is implemented by enumerating all $2^B$ block configurations and sampling according to their exact conditional probabilities. This procedure produces exact samples from the target distribution and serves solely as a zero–sampling-error reference.

### E.5. Evaluation metrics

We evaluate decoding quality using three complementary metrics.

**Reverse KL divergence.**

$$\mathrm{KL}(q\|p) = \mathbb{E}_{x\sim q}[\log q(x) - \log p(x)], \tag{82}$$

estimated via Monte Carlo sampling from the decoding distribution.

**Forward KL divergence.**

$$\mathrm{KL}(p\|q) = \mathbb{E}_{x\sim p}[\log p(x) - \log q(x)], \tag{83}$$

estimated via Monte Carlo sampling using the verified decoder.

**Incoherence rate.** We report the fraction of blocks whose true conditional probability satisfies

$$p(x^{(n)} \mid x^{<n}) \leq \tau, \tag{84}$$

with threshold $\tau = 10^{-8}$.

### E.6. Parameter settings

Unless otherwise stated, we use the following parameter configuration:

- Number of latent states: $K = 2$
- Block size: $B = 8$
- Sequence length: $T = 64$
- Initial distribution: $\pi = (0.5, 0.5)$
- Self-transition probability: $a = 0.9$
- Bernoulli parameters: $\rho = (0.9, 0.1)$
- Parity noise: $\eta \in \{10^{-8}, 10^{-7}, 10^{-6}, 10^{-5}, 10^{-4}\}$
- Incoherence threshold: $\tau = 10^{-8}$

All KL values are estimated using Monte Carlo averages over independently generated sequences.

