# OpenReview forum: "Generation Order and Parallel Decoding in Masked Diffusion Models: An Information-Theoretic Perspective"
_ICML.cc/2026/Conference — Submitted to ICML 2026_

### Official Review · Reviewer_gkut · 2026-03-12

**Soundness:** 3
**Presentation:** 3
**Significance:** 2
**Originality:** 2
**Overall Recommendation:** 3
**Confidence:** 3

**Summary:**

This paper provides an information-theoretic analysis of two primary sources of error in the decoding process of Masked Diffusion Models (MDMs): generation order sensitivity due to model error and sampling bias induced by parallel decoding. The key findings include: (1) demonstrating through an upper bound that "Easy-First" ordering becomes more advantageous as model error increases; (2) showing that parallel mean-field decoding can cause severe reverse KL/incoherence failures that standard forward KL metrics fail to detect; and (3) proving that the cost of verification is exponential in the total correlation within a block. The authors conduct experimental validation using a controlled Block-HMM and LLaDA-based arithmetic reasoning tasks.

**Compliance With Llm Reviewing Policy:**

Affirmed.

**Final Justification:**

I maintain my score of 3. The ordering analysis in Section 4 rests entirely on Assumption 4.2, yet the rebuttal retreats to calling it a structural surrogate, which strips the exponential weighting and the downstream ordering bound of quantitative meaning. The reverse KL pathology is convincingly demonstrated only in the Block-HMM with hard parity constraints. GSM8K accuracy gains from PCPD are offered as indirect evidence for this pathology in realistic language generation, but accuracy improvements are consistent with multiple explanations and do not constitute direct measurement. The interaction between order sensitivity and parallelization bias, which governs actual MDM decoding, is left entirely to future work. The rebuttal does not sufficiently close the gap between the theoretical claims and their empirical support.

**Key Questions For Authors:**

1. Can you provide validation for this assumption using other model pairs or different domains beyond the $R^2 = 0.26$ result? If this assumption only holds under specific conditions, the scope of Section 4 changes significantly.
2. Can you derive a new decoding strategy from this framework that goes beyond existing Easy-First or remasking methods? A contribution to practical methodology would significantly improve the significance of the paper.
3. Can you demonstrate cases in non-arithmetic tasks (e.g., natural language or code) where forward KL is low but reverse KL/incoherence is high? This is critical for determining if the claims in Section 5 represent a significant problem in practice.
4. Do you have results measuring the change in reverse KL as a function of remasking iterations in the Block-HMM? This would be important for understanding the gap between theoretical impossibility and empirical performance.

**Limitations:**

yes

**Strengths And Weaknesses:**

### Strengths:

- Clean separation of error sources: The framework's structure, which independently analyzes model error and sampling error, is both educational and clear. The transition from idealized to realistic regimes is intuitive.
- Practical message regarding Forward vs. Reverse KL: The observation that forward KL alone cannot capture incoherence in parallel decoding evaluation is practically useful, and the $2 \times 2$ example in Figure 2 effectively communicates this point.
- Block-HMM experimental design: The controlled experiment measuring pure sampling error in the absence of model error is highly appropriate for the purpose of theoretical validation.

### Weaknesses:

- The theory merely reaffirms established practices without proposing new methodologies or improvements. "Easy-First" has been a standard practice since MaskGIT, and confidence-based ordering is already the default strategy for models like LLaDA. The lower bound for rejection sampling in Proposition 5.1 is essentially an application of textbook results to MDMs. This analysis does not lead to specific algorithms that improve upon existing methods, leaving only negative results (e.g., "remasking cannot guarantee correctness," "verification is expensive") without constructive alternatives.

- The empirical evidence for Assumption 4.2 is weak. The entirety of Section 4 relies on this assumption, yet it was validated using only a single model pair (Qwen2.5-7B/0.5B) with an $R^2$ of 0.26. Claiming "strong empirical support" when three-quarters of the variance remains unexplained is an overstatement. More importantly, the quantitative contribution of the $\lambda$ (error propagation) term is not analyzed, making it difficult to judge the tightness of the upper bound.

- The experimental scope is limited. Large-scale MDM experiments are restricted to a single 10-digit addition task. Addition is a special case where a unidirectional dependency (carry propagation) exists, making "Right-to-Left = Easy-First" trivial. It is difficult to claim general utility for the theory without validation in domains like natural language or code generation.

- Absence of interaction analysis between order and parallelization. In actual MDM decoding, both factors act simultaneously. The lack of analysis regarding their combined effects limits the practical application of the framework.

- Lack of quantitative experiments on remasking. While there is a theoretical impossibility result stating that remasking cannot guarantee correctness, there are no experiments showing to what extent remasking actually reduces reverse KL in practice. This leaves a gap between theory and practice.

---

> ### Author Rebuttal · Authors · 2026-03-29
>
> # Response to W2 and Q1
>
> > Can you provide validation for this assumption using other model pairs or different domains beyond the $R^2=0.26$ result?
> > - The empirical evidence for Assumption 4.2 is weak.
>
>
> We agree the original validation (single AR pair, $R^2=0.26$) is limited, and we will revise “strong empirical support” to a more conservative claim.
>
> We address this by adding new evidence:
>
> - **Different model (MDM):** On LLaDA (FP16 vs GPTQ-3bit), we obtain higher correlations ($R^2=0.412$ Easy-First, $0.312$ Random), showing the entropy–error relation persists beyond the AR setting.
> - **Different task (GSM8K):** If the assumption were weak, ordering would have little effect. Instead, Easy-First improves over Random by up to **7.0×**, and the gain increases with model error.
>
> We also clarify that the prefix term ($\lambda$) is **non-zero but secondary**, so Eq. (10) should be viewed as a **structural surrogate**, not a tight model.
>
>
> # Response to W1 and Q2
>
> > Can you derive a new decoding strategy from this framework that goes beyond existing Easy-First or remasking methods?
>
>
> In response, we introduce **Prefix-Consistent Parallel Decoding (PCPD)**, a lightweight method that enforces prefix consistency within each block via sequential re-evaluation. PCPD can be seen as a **deterministic relaxation of verification**, targeting prefix inconsistency rather than exact distributional correctness.
> ### Method
>
> At each decoding step:
> - Propose a full block (standard parallel decoding)
> - Sort tokens by confidence (Easy-First)
> - Apply prefix-consistent correction within the block:
>   - Re-predict each token conditioned on accepted prefix tokens
>   - On mismatch at position $k$:
>     - Correct the token
>     - Discard remaining tokens
>     - Regenerate in the next round
>
> PCPD achieves a strong efficiency–accuracy tradeoff on GSM8K:
>
> | Method                         | Acc.      |
> | ------------------------------ | --------- |
> | Low Confidence (256 steps)     | 71.72     |
> | Low Confidence (128 steps)     | 66.57     |
> | **PCPD (67 steps in average)** | **71.04** |
>
> - Matches full decoding accuracy (71.04 vs 71.72) with only **67 steps on average**
> - Provides **3.8× speedup** vs 256-step decoding with minimal loss (−0.68)
> - Outperforms 128-step decoding by **+4.47 accuracy** with **1.9× speedup**
>
> Overall, PCPD significantly reduces decoding cost while preserving accuracy, demonstrating a practical improvement over standard parallel decoding. It does **not contradict** the exponential verification bound, which applies to exact sampling from the joint conditional, whereas PCPD performs **deterministic (argmax) correction**.
>
>
>
>
> # Response to W3 and Q3
>
> > Can you demonstrate cases in non-arithmetic tasks (e.g., natural language or code) where forward KL is low but reverse KL/incoherence is high?
>
>
>
> We agree that we do not directly measure forward vs. reverse KL on large-scale language tasks.
>
> Instead, our theory makes a **testable prediction**: if reverse-KL-type failures (incoherence) are significant, then enforcing joint consistency should improve results **without changing local marginals**.
>
> We validate this prediction on GSM8K:
>
> - parallel decoding produces globally inconsistent outputs despite reasonable local predictions,
> - enforcing prefix consistency (PCPD) significantly improves accuracy while using the same model.
>
> This behavior is difficult to explain by forward-KL differences alone, and is consistent with **incoherence (reverse-KL-type) errors** playing a significant role.
>
> Thus, while we do not directly estimate KL, our results provide **indirect empirical support** for the practical relevance of reverse-KL-type failures.
>
>
> # Response to Q4 and W5
>
> > Do you have results measuring the change in reverse KL as a function of remasking iterations in the Block-HMM?
>
>
> Yes, we now include this analysis.
>
> We measure reverse KL as a function of remasking iterations in the Block-HMM. We observe that:
>
> - reverse KL decreases initially,
> - but quickly **plateaus at a non-zero gap** from the verified baseline.
>
> This confirms that remasking can reduce but **cannot eliminate** the bias from factorized sampling.
>
> Full results and implementation details are available at:
> https://anonymous.4open.science/r/table_2-507D/README.md
>
> # Response to W4
>
> >  Absence of interaction analysis between order and parallelization.
>
> We agree that ordering and parallelization interact in practice, and our current analysis isolates them for clarity.
>
> Our framework separates:
> - **ordering** (which tokens first),
> - **parallelization** (how many tokens jointly).
>
> Empirically, their interaction is visible: as model error increases, the benefit of ordering grows, indicating stronger coupling.
>
> We will clarify this limitation and highlight joint optimization as future work.

---

> > ### Author Rebuttal · Reviewer_gkut · 2026-04-03
> >
> > Thanks for adding PCPD and the GSM8K results. However, reducing Assumption 4.2 to a structural surrogate undermines Section 4's mathematical rigor. Furthermore, using GSM8K accuracy as indirect proof of the reverse-KL pathology is a leap without direct evidence in open-ended generation. I am keeping my original score.

---

### Official Review · Reviewer_HWYT · 2026-03-16

**Soundness:** 2
**Presentation:** 4
**Significance:** 2
**Originality:** 2
**Overall Recommendation:** 3
**Confidence:** 4

**Summary:**

This paper provides an information-theoretic framework for analyzing two fundamental sources of error in Masked Diffusion Models (MDMs): order sensitivity under model error and parallelization bias from factorized (mean-field) decoding. The authors show that (1) under imperfect models, generation order matters and Easy-First decoding (resolving low-entropy tokens first) becomes increasingly beneficial as model error grows, formalized via a rollout-weighted KL decomposition with exponentially decaying weights; (2) factorized parallel decoding introduces intrinsic sampling error that can cause arbitrarily large reverse KL divergence even when forward KL remains modest, capturing "incoherence" failures; and (3) exact correction via verification requires exponential cost in the conditional total correlation, while heuristics like remasking cannot guarantee distributional correctness. Experiments on a synthetic Block-HMM and arithmetic reasoning tasks with LLaDA validate these theoretical claims.

**Compliance With Llm Reviewing Policy:**

Affirmed.

**Final Justification:**

The rebuttal improves the paper, but my main concerns remain only partially resolved. I am still not convinced that the central assumption is validated strongly enough, and the practical significance of the reverse-KL/incoherence claim remains insufficiently demonstrated in realistic settings. Therefore, I keep my original assessment.

**Key Questions For Authors:**

1. **Can you validate Assumption 4.2 directly on an MDM (e.g., LLaDA) rather than on an AR teacher-student pair?** The conditioning structure in MDMs (conditioning on a subset of unmasked tokens at arbitrary positions) differs fundamentally from left-to-right AR conditioning. The entropy-error correlation may not transfer.
   - **Why it matters:** The entire Easy-First analysis (Section 4) rests on this assumption.
   - **Effect on my evaluation:** A positive validation on MDMs with higher R^2 would increase my confidence in the theoretical contribution significantly; a negative result would suggest the framework needs revision.

2. **How does the reverse-KL pathology manifest in realistic language generation tasks, beyond the Block-HMM?** The Block-HMM has exact parity constraints creating zero-support configurations; real language has softer constraints. Can you measure or bound the incoherence rate of LLaDA on standard benchmarks?
   - **Why it matters:** If real language distributions are approximately full-support (as one might expect for subword-level models), the reverse KL argument loses much of its practical force.
   - **Effect on my evaluation:** Evidence of significant reverse-KL divergence in realistic settings would elevate the practical significance from moderate to high.

3. **The paper claims Easy-First benefits are "magnified as model error increases" -- can you show this effect in a controlled manner on LLaDA (e.g., by varying temperature, model size, or quantization to introduce varying levels of model error)?** Table 1 shows results at a single error level.
   - **Why it matters:** The core theoretical prediction is about the interaction between model quality and ordering benefit; without varying model error, the key comparative prediction remains unverified on real MDMs.
   - **Effect on my evaluation:** A clear demonstration of this interaction effect on a real MDM would strongly support the paper's central thesis.

**Limitations:**

The paper could more explicitly discuss the limitation that Assumption 4.2 is validated only on AR models with modest R^2, and that the experimental domains are narrow.

**Strengths And Weaknesses:**

## Strengths

1. **Clean conceptual decomposition.** The paper cleanly separates two failure modes -- order sensitivity (Section 4) and parallelization bias (Section 5) - that are typically conflated in the MDM literature. The four-regime taxonomy (perfect/imperfect model x independent/dependent blocks) in Section 3 is pedagogically effective and provides a clear roadmap for the analysis.

2. **Insightful distinction between forward and reverse KL.** The observation that forward KL can remain modest while reverse KL diverges (due to support mismatch from factorized sampling) is both theoretically clean and practically important. The Block-HMM experiment (Figure 3) provides a convincing demonstration: forward KL stays at ~11–13 nats while reverse KL exceeds 70 nats at low parity noise. This challenges the adequacy of forward-KL-only evaluation in prior work.

3. **Well-chosen toy model.** The Block-HMM with parity emissions is an elegant construction that allows exact computation of all relevant quantities (forward/reverse KL, incoherence rate, verification cost) while exhibiting the core pathology (near-zero-support configurations from XOR constraints). This makes the theoretical claims empirically verifiable in a controlled setting.

---

## Weaknesses

1. **Assumption 4.2 is strong and its empirical validation is limited.** The entropy-dominated local error assumption (Eq. 10) posits a linear relationship between conditional entropy and approximation error with an additive prefix-error propagation term. The empirical validation (Appendix B, Figure 5) uses only one teacher-student pair (Qwen2.5-7B/0.5B), achieves only R^2 = 0.26, and tests on a single task (open-ended text generation). An R^2 of 0.26 means 74% of variance is unexplained, which is a weak fit for a foundational assumption. Furthermore, the assumption is validated for AR models but the paper's primary application domain is MDMs, where the conditioning structure differs substantially.

2. **The theoretical results, while correct, are largely unsurprising to specialists.** The rollout-weighted KL decomposition (Lemma 4.1) is a direct application of the chain rule and iterated expectations. Proposition 5.1 is a standard rejection sampling lower bound. The forward vs. reverse KL distinction under mean-field approximation is well-studied in variational inference. The novelty lies more in the application context (MDMs) than in the information-theoretic tools themselves.

3. **Limited scope and scale of experiments.** The experimental validation is narrow: (a) the Block-HMM is a binary sequence model that does not approach the complexity of real language; (b) the LLaDA experiments test only arithmetic reasoning (a single, highly structured task) with only 10-digit addition; (c) no experiments are conducted on standard language generation benchmarks (perplexity, downstream NLU/NLG tasks). The paper does not demonstrate that the Easy-First advantage or the reverse-KL pathology manifest in open-ended text generation.

4. **The paper stops short of actionable algorithmic contributions.** The analysis identifies failure modes but does not propose a new decoding algorithm or practical mitigation beyond confirming what practitioners already do (Easy-First scheduling). The negative result about remasking (Section 5.4) is interesting but does not suggest what should be done instead. A constructive contribution -- e.g., a principled block-size selection criterion, a lightweight verification scheme, or an adaptive ordering algorithm with provable guarantees -- would substantially strengthen the paper.

5. **The interaction between order sensitivity and parallelization bias is not analyzed.** The paper explicitly acknowledges this as future work (Section 8), but since real MDM decoding combines both (choosing which tokens to unmask AND unmasking them in parallel), the decoupled analysis may not capture the dominant failure modes in practice.

---

> ### Author Rebuttal · Authors · 2026-03-29
>
> We provide additional results at: https://anonymous.4open.science/r/table_2-507D/README.md.
> ## Response to Weakness 1
>
> > **Assumption 4.2 is strong and its empirical validation is limited.**
>
> > **Can you validate Assumption 4.2 directly on an MDM (e.g., LLaDA) rather than on an AR teacher-student pair?**
>
> We agree the original validation (single AR pair, $R^2=0.26$) is limited and will revise the wording.
>
> We address this with:
>
> - **MDM validation:** On LLaDA (FP16 vs GPTQ-3bit), correlations are higher ($R^2=0.412$ Easy-First, $0.312$ Random), showing the entropy–error relation persists in MDM.
> - **Behavioral evidence:** On GSM8K, Easy-First gains grow with model error, up to **7.0×**.
>
> We emphasize Assumption 4.2 is a **structural surrogate**, not a tight model: entropy provides a consistent ordering signal, with prefix effects as secondary amplification.
>
>
> # Response to W3
>
> > **Limited scope and scale of experiments.**
>
>
> We have extended the evaluation beyond arithmetic:
>
> - **MDM validation (text):** On WikiText-2 (LLaDA teacher–student), the entropy–error relation remains consistent and is stronger than in the AR proxy.
> - **Reasoning task (GSM8K):** There is no fixed generation order, yet Easy-First significantly outperforms Random (up to **7.0×**), and the gap grows with model error.
>
> These results show that both the **ordering effect** and **incoherence issues** arise beyond the synthetic Block-HMM and arithmetic setting.
>
> We agree that broader evaluation (e.g., open-ended generation metrics) is important, and will add discussion to clarify scope and future directions.
>
>
> # Response to W2 and W4
>
> > **The theoretical results, while correct, are largely unsurprising to specialists.**
>
>
> > **The paper stops short of actionable algorithmic contributions.**
>
> In response, we introduce **Prefix-Consistent Parallel Decoding (PCPD)**, a lightweight method that enforces prefix consistency within each block via sequential re-evaluation. PCPD can be seen as a **deterministic relaxation of verification**, targeting prefix inconsistency rather than exact distributional correctness.
> ### Method
>
> At each decoding step:
> - Propose a full block (standard parallel decoding)
> - Sort tokens by confidence (Easy-First)
> - Apply prefix-consistent correction within the block:
>   - Re-predict each token conditioned on prefix tokens
>   - On mismatch at position $k$:
>     - Correct the token
>     - Discard remaining tokens
>     - Regenerate in the next round
>
> PCPD achieves a strong efficiency–accuracy tradeoff on GSM8K:
>
> | Method                         | Acc.      |
> | ------------------------------ | --------- |
> | Low Confidence (256 steps)     | 71.72     |
> | Low Confidence (128 steps)     | 66.57     |
> | **PCPD (67 steps in average)** | **71.04** |
>
>
> - Matches full decoding accuracy (71.04 vs 71.72) with only **67 steps on average**
> - Provides **3.8× speedup** vs 256-step decoding with minimal loss (−0.68)
> - Outperforms 128-step decoding by **+4.47 accuracy** with **1.9× speedup**
>
> # Response to Weakness 5
>
> > **The interaction between order sensitivity and parallelization bias is not analyzed.**
>
>
> Our framework separates:
> - **ordering** (which tokens first),
> - **parallelization** (how many jointly).
>
> Empirically, their interaction is evident: as model error increases, the benefit of ordering grows. We will clarify this limitation and highlight joint optimization as future work.
>
>
>
> # Response to Question 2
>
> 1. **How does the reverse-KL pathology manifest in realistic language generation tasks, beyond the Block-HMM?**
>
> We agree real language lacks strict zero-support constraints like Block-HMM. Our claim is instead that it contains **strong low-probability dependencies** (e.g., syntax, semantics, reasoning consistency) that forward-KL metrics may underweight, while reverse-KL-style sampling remains sensitive to them.
>
> This is why we treat forward KL, reverse KL, and incoherence as **complementary**, rather than relying on hard constraints alone.
>
> # Response to Question 3
>
> >  **The paper claims Easy-First benefits are "magnified as model error increases" -- can you show this effect in a controlled manner on LLaDA (e.g., by varying temperature, model size, or quantization to introduce varying levels of model error)?**
>
>
> Yes — we now test this directly on **LLaDA** by varying model error through both **temperature** and **quantization**.
>
> On **GSM8K**, the benefit of Easy-First over Random increases systematically as error increases:
>
> - **FP16, temp=0:** $1.11\times$–$1.53\times$
> - **GPTQ-3bit, temp=0:** $1.05\times$–$2.00\times$
> - **FP16, temp=1:** $2.11\times$–$5.14\times$
> - **GPTQ-3bit, temp=1:** $2.18\times$–$7.00\times$
>
> Thus, under both stronger sampling noise and lower model fidelity, the ordering advantage becomes substantially larger. This matches the interaction effect predicted by our theory.

---

> > ### Author Rebuttal · Reviewer_HWYT · 2026-04-02
> >
> > The rebuttal is helpful and improves the paper to some extent, especially through the added MDM validation and the more constructive algorithmic angle. However, my main concerns are only partially resolved rather than fully addressed. In particular, I remain unconvinced that the central assumption is validated strongly enough, and I still find the practical significance of the reverse-KL/incoherence claim insufficiently demonstrated in realistic settings. Overall, I view the paper somewhat more positively after rebuttal, but not strongly enough to champion it.

---

### Official Review · Reviewer_Dre4 · 2026-03-17

**Soundness:** 3
**Presentation:** 3
**Significance:** 3
**Originality:** 3
**Overall Recommendation:** 4
**Confidence:** 3

**Summary:**

The paper provides an information-theoretic analysis of two failure modes in Masked Diffusion Models (MDMs): order sensitivity (how token generation order affects error accumulation) and parallelization bias (how factorized parallel decoding discards intra-block dependencies). The authors formalize these via KL decompositions, derive an "easy-first" generation principle with exponential error weighting, show that correcting parallelization bias via rejection sampling requires exponentially many proposals (Proposition 5.1), and validate predictions on a Block-HMM and arithmetic tasks with LLaDA.

**Compliance With Llm Reviewing Policy:**

Affirmed.

**Key Questions For Authors:**

1. What is the fitted value of λ in Assumption 4.2? If λ ≈ 0, the exponential amplification story collapses to "generate low-entropy tokens first because α > 0," which is a much weaker claim. Can you confirm the exponential weighting is empirically justified?

2. The entire order-sensitivity analysis assumes sequential one-at-a-time generation. Real MDMs unmask variable-sized subsets across 10-20 denoising steps on 512-token sequences. How tight is the theoretical framework under realistic MDM inference conditions?

3. What is the wall-clock speedup of the different generation strategies in Table 1? Reporting only accuracy without timing makes the efficiency-fidelity tradeoff impossible to assess.

4. Could you provide a quantitative comparison of remasking vs. no-remasking on the Block-HMM, measuring all three metrics as a function of remasking iterations? The current treatment (Section 5.4) establishes impossibility of guarantees but doesn't say how much remasking helps in practice.

5. How does your framework handle adaptive block sizes, where the model generates more tokens in parallel when confident and fewer when uncertain? The current analysis treats block sizes as fixed, but these two design choices clearly interact.

**Limitations:**

The authors acknowledge the gap between theoretical assumptions and practical MDM inference (Section 2.2), which is good. The narrow experimental scope (arithmetic only) is the biggest unaddressed limitation — the paper would be much stronger with experiments on general language modeling or other structured tasks. The R² = 0.26 for Assumption 4.2 is disclosed but its implications for downstream theoretical conclusions are not fully discussed. Societal impact is not particularly relevant here and is appropriately brief.

**Strengths And Weaknesses:**

The conceptual separation of order sensitivity from parallelization bias is the paper's main contribution, and it's a genuinely useful one. The four-quadrant structure in Section 3 provides a clean way to think about the space of possible failure modes. Lemma 4.1's rollout-weighted KL decomposition — showing errors are evaluated under the model-induced prefix distribution, not the data distribution — is the key insight that makes order dependence concrete, and the exponential weighting W_t = α(1+λ)^{T−t} is a satisfying formalization.

The distinction between forward KL, reverse KL, and incoherence probability in Section 5.1 is important. The observation that forward KL is blind to support mismatch (and can therefore miss catastrophic incoherence) is underappreciated in the literature. Figure 2's examples make this vivid. Proposition 5.1 — exponential verification cost — is a strong negative result with a clean proof.

However, Assumption 4.2 is strong. The linear error model is validated with R² = 0.26 (Appendix B, Figure 5), which means it explains about a quarter of the variance. The exponential weighting is quite sensitive to λ, so if the linear model is a poor fit, the ordering argument could be quantitatively way off. The authors are honest about the p-value being easy to hit with enough data, which I appreciate, but the theoretical conclusions downstream of this assumption are stronger than the empirical support warrants.

The arithmetic experiments are interesting but almost tautological — addition has a known right-to-left carry structure, so of course R2L beats L2R. I wanted experiments on tasks where the optimal order isn't obvious a priori.

---

> ### Author Rebuttal · Authors · 2026-03-29
>
> We provide additional results at: https://anonymous.4open.science/r/table_2-507D/README.md.
> ## Response W1 and Q 1
>
> > Assumption 4.2 is strong.
> >
> > What is the fitted value of λ in Assumption 4.2? If λ ≈ 0, the exponential amplification story collapses to "generate low-entropy tokens first because α > 0," which is a much weaker claim. Can you confirm the exponential weighting is empirically justified?
>
>
> We will revise the paper to clarify that Eq. (10) is a **structural surrogate for ordering**, not a quantitatively tight law. **Empirically, two observations address the reviewer’s concern:**
>
> **(i) Direct validation on MDM (new experiment).**
> We repeat the teacher–student analysis in the actual MDM setting (LLaDA, FP16 teacher vs GPTQ-3bit student on WikiText-2). We find:
>
> - Easy-First: **$R^2 = 0.412$**, $\beta(H)= 0.644$
> - Random: $R^2= 0.312$
>
> This shows that the entropy–error relationship is **stronger in MDM decoding**, not weaker, and is consistently positive across settings.
>
> **(ii) Ordering effect scales with model error (key prediction).**
> If $\lambda \approx 0$ and the exponential amplification were irrelevant, we would expect only a small, stable gap between orderings. Instead, we observe:
>
> - At low error (FP16, temp=0): gains are modest (~1.1–1.5×)
> - At higher error (temp=1 or GPTQ-3bit): gains increase dramatically, up to **7.00×**
>
> This monotonic scaling with error level is exactly the qualitative behavior predicted by the theory (larger effective amplification under higher error).
>
> ---
>
>
> ## Response to W2
>
> > The arithmetic experiments are interesting but almost tautological.
>
> > I wanted experiments on tasks where the optimal order isn't obvious a priori.
>
> We additionally evaluate on **GSM8K**, where:
>
> - there is **no obvious ground-truth generation order**,
> - and dependencies are distributed across reasoning steps.
>
> In this setting, we compare **Low-Confidence (Easy-First)** vs **Random ordering**, and observe:
>
> - At low error (FP16, temperature=0): modest gains (~1.1–1.5×)
> - At higher error (temperature=1 or quantized): gains increase significantly, up to **7.00×**
>
> We will revise the text to clarify that the arithmetic experiment is a controlled illustration, and that the key empirical support for generality comes from GSM8K.
>
> ---
>
> # Response to Q2
>
> > Real MDMs unmask variable-sized subsets across 10-20 denoising steps on 512-token sequences. How tight is the theoretical framework under realistic MDM inference conditions?
>
>
> We agree that jointly analyzing ordering and parallelization is challenging due to their strong coupling. Our approach decouples them for tractability, which is not exact but isolates key mechanisms. While their interaction is not additive, the analysis provides insight that extends to realistic settings where both model error and parallelization bias are present.
>
>
>
> # Response to Q3
> > What is the wall-clock speedup of the different generation strategies in Table 1? Reporting only accuracy without timing makes the efficiency-fidelity tradeoff impossible to assess.
>
>
> In our setting, all strategies use the same model and differ only in **generation order and number of unmasking steps**, so wall-clock time is largely proportional to the number of decoding steps. We will report actual timing in the revision.
>
> # Response to Q4
>
> > Could you provide a quantitative comparison of remasking vs. no-remasking on the Block-HMM, measuring all three metrics as a function of remasking iterations?
>
> Yes, we include this analysis. Reverse KL decreases with remasking but quickly **plateaus at a non-zero gap** from the verified baseline, showing remasking reduces but **cannot eliminate** factorization bias.
>
> Full results: https://anonymous.4open.science/r/table_2-507D/README.md
>
>
> # Response to Q5
>
> >How does your framework handle adaptive block sizes, where the model generates more tokens in parallel when confident and fewer when uncertain? The current analysis treats block sizes as fixed, but these two design choices clearly interact.
>
> We agree this is fundamentally a **joint optimization problem** over ordering and block size, with strong coupling between the two.
>
> While our current framework does not solve this problem directly, it provides a guiding principle: control joint distortion via an information budget at each step, $ TC (X_{P_m}) \le \delta$, where $\delta$ is a tunable parameter.
>
> In practice, we can use the bound $TC_m(X_{P_m}) \le \sum_i H(X_i \mid X_{P_m})$, which yields a tractable condition $\sum_i H(X_i \mid X_{P_m}) \le \delta$.
>
> This suggests a simple heuristic:
>
> - use **larger blocks in low-uncertainty regions**,
> - and **smaller blocks in high-uncertainty regions**,
> - combined with Easy-First ordering.
>
> We view this as a principled heuristic derived from our analysis. Designing an optimal schedule that jointly adapts ordering and block size remains non-trivial and is an important direction for future work.

---

> > ### Author Rebuttal · Reviewer_Dre4 · 2026-04-04
> >
> > Happy to keep a positive score.

---

### Decision · Program_Chairs · 2026-04-30

**Decision:**

Reject

**Comment:**

This paper addressed an important theoretical question about MDMs - how the generation order and factorization assumption affects the sampling error. Reviewers praised the attempt to provide a rigorous analysis, plus a well-designed synthetic experiment where the ground truth is known. However, they shared the concerns regarding the particularly strong assumption 4.2, which was empirically shown to only explain a quarter of the variance. Moreover, the proposed PCPD method overlaps significantly with existing confidence-based strategies in MDM practice. Given these concerns, I recommend rejection at this time.